



# On the role of trend and variability of hydroxyl radical (OH) in the global methane budget

Yuanhong Zhao[1], Marielle Saunois[1], Philippe Bousquet[1], Xin Lin[1*], Antoine Berchet[1], Michaela I. Hegglin[2], Josep G. Canadell[3], Robert B. Jackson [4], Makoto Deushi[5], Patrick Jöckel[6], Douglas Kinnison[7], Ole Kirner[8], Sarah Strode[9, 10], Simone Tilmes[11], Edward J. Dlugokencky[12], and Bo Zheng[1]

[1] Laboratoire des Sciences du Climat et de l'Environnement, LSCE-IPSL (CEA-CNRS-UVSQ), Université Paris-Saclay, 91191 Gif-sur-Yvette, France

[2] Department of Meteorology, University of Reading, Earley Gate, Reading RG6 6BB, United Kingdom

[3] Global Carbon Project, CSIRO Oceans and Atmosphere, Canberra, Australian Capital Territory 2601,
Australia

[4] Earth System Science Department, Woods Institute for the Environment, and Precourt Institute for Energy, Stanford University, Stanford, CA 94305, USA

[5] Meteorological Research Institute, 1-1 Nagamine, Tsukuba, Ibaraki, 305-0052, Japan

[6]Deutsches Zentrum für Luft- und Raumfahrt (DLR), Institut für Physik der Atmosphäre,
Oberpfaffenhofen, Germany

[7]Atmospheric Chemistry Observations and Modeling Laboratory, National Center for Atmospheric Research, 3090 Center Green Drive, Boulder, CO, 80301, USA

[8] Steinbuch Centre for Computing, Karlsruhe Institute of Technology, Karlsruhe, Germany

[9] NASA Goddard Space Flight Center, Greenbelt, MD, USA
[10] Universities Space Research Association (USRA), GESTAR, Columbia, MD, USA

[11] National Center for Atmospheric Research, Boulder, CO, USA

[12] Global Monitoring Division, NOAA Earth System Research Laboratory, Boulder, CO,

* Now at: Climate and Space Sciences and Engineering, University of Michigan, Ann Arbor, MI 48109, USA

*Correspondence to*: Yuanhong Zhao (yuanhong.zhao@lsce.ipsl.fr) and Bo Zheng

(bo.zheng@lsce.ipsl.fr)



## Abstract

Decadal trends and interannual variations in the hydroxyl radical (OH), while poorly constrained at present, are critical for understanding the observed evolution of atmospheric methane ($CH_4$). Through analyzing the OH fields simulated by the model ensemble of the Chemistry-Climate Model Initiative (CCMI), we find (1) the negative OH anomalies during the El Niño years mainly corresponding to the enhanced carbon monoxide (CO) emissions from biomass burning and (2) a positive OH trend during 1980-2010 dominated by the elevated primary production and the reduced loss of OH due to decreasing CO after 2000. Both two-box model inversions and variational 4D inversions suggest that ignoring the negative anomaly of OH during the El Niño years leads to a large overestimation of the increase in global $CH_4$ emissions by up to 10Tg $yr^{-1}$ to match the observed $CH_4$ increase over these years. Not accounting for the increasing OH trends given by the CCMI models leads to an underestimation of the $CH_4$ emission increase by ~23Tg $yr^{-1}$ from 1986 to 2010. The variational inversion estimated $CH_4$ emissions show that the tropical regions contribute most to the uncertainties related to OH. This study highlights the significant impact of climate and chemical feedbacks related to OH on the top-down estimates of the global $CH_4$ budget.



## 1 Introduction

Methane ($CH_4$) in the Earth's atmosphere is a major anthropogenic greenhouse gas that has resulted in a 0.62 W m$^{-2}$ additional radiative forcing from 1750 to 2011 (Etminan et al., 2016). The tropospheric $CH_4$ mixing ratio has more than doubled between pre-industrial and the present day, which is unambiguously attributed to the increasing anthropogenic $CH_4$ emissions (Etheridge et al., 1998). Although the centennial and inter-decadal trends and the drivers of $CH_4$ growth are fairly clear, it is still challenging to understand the trends and the associated interannual variations on a time scale of 1-30 years. For example, the mysterious stagnation in $CH_4$ mixing ratios during 2000-2007 (Dlugokencky, NOAA/ESRL, 2019) is still under debate, highlighting the need for closing gaps in the global $CH_4$ budget on decadal time scales (e.g. Turner et al., 2019).

One of the barriers to understanding atmospheric $CH_4$ changes is the $CH_4$ sink, which is mainly the chemical reaction with the hydroxyl radical (OH) (Saunois et al., 2016; 2017; 2019; Zhao et al., 2020) that determines the tropospheric $CH_4$ lifetime. The burden of atmospheric OH is determined by complex and coupled atmospheric chemical cycles influenced by anthropogenic and natural emissions of multiple atmospheric reactive species, and also by climate change (Murray et al., 2013; Turner et al., 2018, Nicely et al., 2018), making it difficult to diagnose OH temporal changes from a single process. The OH source mainly include the primary production from the reaction of excited oxygen atoms ($O(^1D)$) with water vapor ($H_2O$) and the secondary production mainly from the reaction of nitrogen oxide (NO) or ozone ($O_3$) with hydroperoxyl radical ($HO_2$) or organic peroxy radicals ($RO_2$). The OH sinks mainly include the reaction of OH with carbon monoxide (CO), $CH_4$, or non-methane volatile organic compounds (NMVOCs).

Based on inversions of *1-1-1 trichloroethane* (methyl chloroform, MCF) atmospheric observations, some previous studies have attributed part of the observed $CH_4$ changes to the temporal variation in OH



concentrations ([OH]) but report large uncertainties in their estimates (McNorton et al 2016; Rigby et al.

2008, 2017; Turner et al., 2017). Such proxy approaches based on MCF inversions also have limitations

in their accuracy, both due to uncertainties in MCF emissions before the 1990s, and the weakening of

MCF gradients after the 1990s (Krol et al., 2003, Bousquet et al., 2005; Montzka et al., 2011; Prather and

Holmes, 2017).


The OH variations have been explored with atmospheric chemistry models in terms of climate change

(Nicely et al., 2018), anthropogenic emissions (Gaubert et al. 2017), and lightning $NO_x$ emissions (Murray

et al., 2013; Turner et al., 2018). The El Niño-Southern Oscillation (ENSO) has proven to influence [OH]

by perturbing CO emissions from biomass burning (Rowlinson et al. 2019) and $NO_x$ emissions from

lightning (Turner et al., 2018), but the detailed mechanisms behind present OH variations and their impact

on the $CH_4$ budget remain poorly understood. Nguyen et al. (2020) estimated the impact of the chemical

feedback induced by CO and $CH_4$ changes on the top-down estimates of $CH_4$ emissions using a box model

approach. However, they account neither for the heterogeneous distribution of atmospheric reactive

species in space nor for the chemical feedback related to OH production processes that vary over time.

Understanding the influences of the chemical feedback related to OH on $CH_4$ emissions as estimated by

atmospheric inversions is urgently needed and can benefit from better incorporating 3D simulations from

atmospheric chemistry models.

Here we continue our former studies (Zhao et al., 2019; 2020), in which we have quantified the impact of

OH on top-down estimates of $CH_4$ emissions during the 2000s. This work aims to better understand the

production and loss processes of OH and quantitatively assess their influence on the temporal changes of

$CH_4$ lifetime and the global $CH_4$ budget on decadal-scale from 1980 to 2010. We first analyze the trends

and year-to-year variations of nine independent OH fields covering the period of 1980-2010 simulated by

the phase 1 of the International Global Atmospheric Chemistry (IGAC)/Stratosphere-troposphere





Processes and their Role in Climate (SPARC) Chemistry-Climate Model Initiative (CCMI) models (Morgenstern et al., 2017) and then assess the contribution of different chemical processes to the OH budget by estimating the main OH production and loss processes. We finally estimate the impact of OH year-to-year variations and trends on the top-down estimation of global $CH_4$ emissions. Two-box model inversions and the variational 4D inversions are both used to assess how the nonlinear chemical feedback

related to OH influences our understanding of the trends and drivers of the global $CH_4$ budget.

## 2 Method

### 2.1 CCMI OH fields

In this study, we analyze the OH fields simulated by five models (CESM1-CAM4Chem, CESM1-

WACCM, EMAC-L90MA, GEOSCCM, MRI-ESM1r1), which include detailed tropospheric ozone chemistry and multiple primary VOC emissions. All five models conducted the REF-C1 experiments (free-running simulations driven by state-of-the-art historical forcings including sea surface temperature and sea ice concentrations) for 1960-2010, and four of them (excluding GEOSCCM) conducted the REFC-1SD experiments (similar to REF-C1 but nudged to the reanalysis meteorology data) for 1980-

2010. Thus, we have nine OH fields generated by models with different chemistry, physics, and dynamics covering the period 1980-2010. A detailed description of these CCMI models, experiments and characteristics of the OH fields can be found in Morgenstern et al. (2017) and Zhao et al. (2019).

To eliminate the influence of different magnitudes of global OH burden simulated by those models, we

scale all OH fields to the same $CH_4$ loss for the year 2000 based on the reaction with OH used in the TRANSCOM inter-comparison exercise (Patra et al., 2011). The inferred global mean scaling factors are calculated for the year 2000 and for each OH field and then applied to the whole period. The production $(O(^1D)+H_2O, NO+HO_2, O_3+HO_2)$ and loss processes (removal of OH by CO, $CH_4$, formaldehyde ($CH_2O$), and isoprene) for each OH field are estimated using the CCMI database (Section S1). For each OH fields,

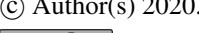



we separate trends and year-to-year variations of the global tropospheric mean OH concentration

($[OH]_{GM-CH4}$, [OH] weighted by reaction rate of OH with $CH_4$ multiplied by dry air mass, Lawrence et al.,

2001) as well as of its production and loss rates.

## 2.2 Atmospheric inversion systems

To evaluate the influences of OH temporal variations on the top-down estimation of $CH_4$ emissions, we

have conducted Bayesian atmospheric inversions using: 1) a two-box model similar to that described by

Turner et al. (2017) and 2) a 4D variational inversion system based on the version LMDz5B of the LMDz

atmospheric transport model under the PYVAR-SACS framework (Chevallier et al., 2007; Pison et al.,

2009) as described by Locatelli et al. (2015) and Zhao et al. (2020). The two-box model inversions allow

us to easily conduct multiple long-term global scale inversions (1984-2012) with each of the nine OH

fields to estimate the global $CH_4$ emission variations caused by various OH fields. The variational

inversions allow us to better represent the atmospheric transport and to address regional $CH_4$ emission

distributions. Thus, we have conducted both, two-box model inversions with each of the nine OH fields,

and variational inversions with the multi-model mean OH field (average of the nine OH fields).


Both the box model and the variational inversions optimize the $CH_4$ emissions and initial mixing ratios

by assimilating the observation data from the Earth System Research Laboratory of the US National

Oceanic and Atmospheric Administration (NOAA/ESRL, Dlugokencky et al. (1994)). The OH

concentrations are prescribed and not optimized in both inversion systems. A detailed description of the

two-box model, the LMDz atmospheric transport model, and the variational inversion method used here

are provided in the supplementary material (Section S2).

## 2.3 Ensemble of different inversions

We have designed an ensemble of inversion experiments as listed in Table 1 using the two-box model

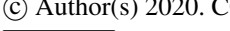

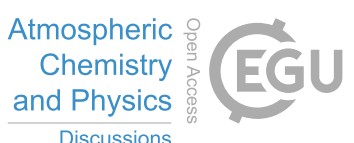

with each OH field. Here, Inv_OH_std uses the aforementioned scaled OH fields; Inv_OH_cli uses a climatology of each OH field, which is constant over the years and correspond to an average over 1980-2010; Inv_OH_var stands for the detrended OH (only keeping the year-to-year variations); Inv_OH_trend uses the OH without the year-to-year variability (retaining only the trend). By comparing Inv_OH_cli with Inv_OH_std, Inv_OH_var, and Inv_OH_trend, it is possible to assess the influence of total OH

temporal changes, year-to-year variations, and OH trends on the overall CH$_4$ changes, respectively. The box model inversions are conducted from 1984 to 2012 (2010 OH fields are used for 2011 and 2012). The first and last two years are treated as spin-up and spin-down, and we only analyze the inversion results over 1986-2010.

We have conducted two variational inversions, Inv_OH_std and Inv_OH_cli, using the multi-model mean OH field to test the influence of OH temporal variations on the top-down estimates of global to regional CH$_4$ emissions. The LMDz inversions are conducted for four time periods (1994-1997, 1996-1999, 2000-2004, and 2006-2010; first and last years as spin-up and spin-down; Sect.3.4). The four time periods are chosen to represent the transition from La Niña (1995-1996) to El Niño (1997-1998) years and the years

of stagnated (2001-2003) and renewed growth (2007-2009) of observed CH$_4$.

## 3 Results

### 3.1 Decadal OH trends and year-to-year variability

All CCMI models simulate positive OH trends from 1980 to 2010 after removing the year-to-year

variability (Fig.1, top panel), consistent with previous analyses of CCMI OH fields (Zhao et al., 2019; Nicely et al., 2020) and model results of the Aerosol and Chemistry Model Intercomparison Project (Stevenson et al., 2020). The multi-model mean [OH]$_{GM-CH4}$ increased by $0.7 \times 10^5$molec cm$^{-3}$ from 1980 to 2010. The growth rates in [OH]$_{GM-CH4}$ are estimated as ~$0.03 \times 10^5$molec cm$^{-3}$ yr$^{-1}$ (0.3% yr$^{-1}$) during



the early 1980s, ~$0.01 \times 10^5$molec cm$^{-3}$ yr$^{-1}$ (0.1% yr$^{-1}$) between the mid-1980s and the late-1990s, and 0.03-0.05$\times 10^5$molec cm$^{-3}$ yr$^{-1}$ (0.3%-0.5% yr$^{-1}$) since the 2000s. This continuously increases in [OH] is different from the results based on the MCF inversions using the two-box model approach (Turner et al., 2017; Rigby et al., 2017), which yield increases in [OH] from the 1990s to the early 2000s and a decreases in OH afterward.

The ensemble of the anomaly of detrended [OH]$_{GM-CH4}$ (middle panel of Fig.1) shows a strong anti-correlation (r = -0.50) with the bi-monthly Multivariate ENSO Index Version 2 (MEI, the bottom panel of Fig.1 and Section S3) (Zhang et al., 2019), with higher [OH]$_{GM-CH4}$ during La Niña and lower [OH]$_{GM-CH4}$ during El Niño. From 1980 to 2010, the CCMI model simulations show several negative [OH]$_{GM-CH4}$ anomalies, the three largest reaching as high as -0.36±0.23$\times 10^5$molec cm$^{-3}$ (-3.5±2.2%) during 1982-1983, -0.37±0.17$\times 10^5$molec cm$^{-3}$ (-3.6±1.7%) during 1991-1992, and -0.49±0.37$\times 10^5$molec cm$^{-3}$ (-4.7±3.5%) during 1997-1998. The negative [OH]$_{GM-CH4}$ anomalies during 1982-1983 and 1997-1998 correspond to the two strongest El Niño events (MEI>2.5). During 1991-1992, the negative [OH]$_{GM-CH4}$ anomaly corresponds to both, the weaker El Niño event (MEI up to 2.0), and the eruption of Mount Pinatubo. During other weak El Niño events (1986-1987, 2002-2003, 2004-2005, and 2006-2007), the multi-model mean [OH]$_{GM-CH4}$ shows smaller negative anomalies of 1-2%. Only the negative OH anomaly during 2006-2007 (2.2±1.0%) is simulated by all models. The negative anomalies are consistent with an up to 9% reduction of [OH] during 1997-1998 simulated by TOMCAT-GLOMAP as shown by Rowlinson et al. (2019), as well as a 5% reduction of [OH] over tropical regions during 1991-1993 constrained by MCF observations (Bousquet et al, 2006). During La Niña events, the [OH]$_{GM-CH4}$ shows ~2% positive anomalies, resulting in more than a 6% increase in OH (max−min) during 1983-1985, 1992-1994, and 1998-2000.

The negative [OH]$_{GM-CH4}$ anomalies during strong El Niño events correspond to the highest growth rates



of the $CH_4$ mixing ratio from the surface observations (*Dlugokencky, NOAA/ESRL*), which are

$14.0\pm0.6$ppbv $yr^{-1}$ in 1991, and $12.1\pm0.8$ppbv $yr^{-1}$ in 1998 (Fig.S1). The positive anomalies of $[OH]_{GM-CH_4}$ during La Niña events correspond to a much smaller $CH_4$ growth (e.g. $3.8\pm0.6$ppbv $yr^{-1}$ in 1993 and $2.3\pm0.8$ppbv $yr^{-1}$ in 1999) (Fig. S1).

## 3.2 Factors controlling OH trends and year-to-year variability

The changes in tropospheric [OH] are due to changes in the balance of production and loss processes. Here we assess the drivers of OH year-to-year variations and trend by calculating the OH production and loss processes listed in Table 2 following Murray et al. (2013; 2014). The multi-model calculated OH production/loss in the troposphere averaged over 1980-2010 is $209.3\pm11.9$Tmol $yr^{-1}$, similar to that ~200Tmol $yr^{-1}$ reported by Murray et al. (2014). Of the total OH production, 46% ($96.2\pm1.9$Tg $yr^{-1}$) are

from primary production ($O(^1D)$ +$H_2O$). Two main secondary productions, $NO+NO_2$, and $O_3+HO_2$ account for 30% ($62.6\pm4.1$Tmol $yr^{-1}$) and 13% ($26.2\pm1.9$Tmol $yr^{-1}$), respectively. For the OH loss, reactions with CO and $CH_4$ account for 39% ($82.3\pm3.8$Tmol $yr^{-1}$) and 15% ($32.4\pm1.4$Tmol $yr^{-1}$), respectively. We have also calculated the OH loss by reactions with isoprene ($C_5H_8$) and formaldehyde ($CH_2O$), which both remove 6% of OH, reflecting the influences of NMVOCs from natural and

anthropogenic sources, respectively. Besides, there are 12% of OH production and 33% of OH loss not analyzed here due to lack of data in the CCMI model outputs (e.g. OH loss due to reaction with NMVOCs included in difference models).

Fig.2 shows the changes in the trends of OH production and loss processes (year-to-year variations are

removed) with respect to the year 1980. The OH primary production ($O (^1D)+H_2O$) shows a large increase of $10.1\pm1.1$Tmol $yr^{-1}$ from 1980 to 2010, as the dominant driver of the positive OH trend. The increase in OH primary production is due to an increase in both tropospheric $O_3$ burden (producing $O(^1D)$) and water vapor (Zhao et al., 2019; Nicely et al., 2020). The OH loss from CO increased by $7.3\pm0.7$Tmol $yr^{-1}$





[1] from 1980 to 2001 but then decreased by $4.2 \pm 2.2$Tmol yr$^{-1}$ from 2001 to 2010. We find that the decrease

in OH loss by CO can explain the accelerated OH increase after 2000, despite a stagnated OH primary

production and a slight decrease of the OH secondary production. The OH loss by $CH_4$, which shows a

continuous increase of $6.2 \pm 0.5$Tmol yr$^{-1}$ from 1980 to 2010, buffers the increase in OH production by NO

($5.3 \pm 1.1$Tmol yr$^{-1}$). The OH production by $O_3 + HO_2$, as well as OH loss by $CH_2O$ and isoprene, show

smaller changes of $2.1 \pm 1.1$Tmol yr$^{-1}$, $1.9 \pm 0.3$Tmol yr$^{-1}$, and $1.1 \pm 0.6$Tmol yr$^{-1}$, respectively, during 1980-

2010. By comparing the magnitude of the production and loss processes, we conclude that an enhanced

OH primary production and changes in OH loss by CO are the most important factors leading to the

increased OH trend inferred by CCMI models from 1980 to 2010.

Fig.3 and Fig.S2 show the year-to-year variations of the global total OH production and loss due to several

processes (calculated after trends have been removed). Year-to-year variations of global [OH] are mainly

determined by the primary ($O(^1D) + H_2O$) and secondary production ($NO + HO_2$; $O_3 + HO_2$) and by OH loss

due to CO (Fig.3). Other OH loss processes, including reactions with $CH_4$, $CH_2O$, and isoprene, show

much smaller year-to-year variations but larger uncertainties (Fig.S2), revealing a larger model spread for

these processes.


As shown in Fig.3, negative anomalies of [OH] during El Niño events are dominated by increased OH

loss through the reaction with CO in response to enhanced biomass burning (Fig.S3), similar to the

conclusions of Rowlinson et al. (2019) and Nicely et al. (2020). During the strong El Niño events in 1982-

1983, 1991-1992, and 1997-1998, the OH loss by CO increased by up to $3.4 \pm 0.4$Tmol yr$^{-1}$, $4.5 \pm 0.6$Tmol

yr$^{-1}$, and $7.6 \pm 0.5$Tmol yr$^{-1}$, respectively, compared to the mean value of 1980-2010. The increase of OH

loss by CO can be partly offset by an increase in OH production. Indeed, in 1998, the OH primary

production ($O(^1D) + H_2O$), OH produced by $NO + RO_2$, and $O_3 + RO_2$ increased by $3.2 \pm 0.7$Tmol yr$^{-1}$,

$2.7 \pm 0.5$Tmol yr$^{-1}$, and $1.6 \pm 0.3$Tmol yr$^{-1}$, respectively, offsetting most of the OH loss increase. The



increase in OH primary production is mainly due to an increase in tropospheric water vapor and $O_3$ burden during El Niño events (Fig.S3 and S12 in Nicely et al. (2020)), while the increase in OH secondary production is caused by enhanced $NO_x$ emissions (Fig.S3) and $O_3$ formation (Nicely et al. (2020) related to biomass burning as well as more $HO_2$ formation by CO+OH. As a result, the OH year-to-year variations found here are much smaller than those estimated by Nguyen et al. (2020), who mainly considered the response of OH to CO. The positive anomaly OH primary production ($0.2 \pm 0.5$ Tmol yr$^{-1}$) is not significant during 1991-1992 El Niño event, maybe due to reduction of tropospheric water vapor (Fig.S3 in Nicely et al. (2020)) after the eruption of Mount Pinatubo (Soden et al., 2020). Thus, the negative [OH] anomaly during the weak El Niño event in 1991-1992 is potentially being enhanced by the eruption of Mount Pinatubo. Previous studies have shown that $NO_x$ emissions from lightning can contribute to the OH interannual variability (Murray et al., 2013; Turner et al. 2018). In addition, soil $NO_x$ emissions depend on temperature and soil humidity (Yienger and Levy, 1995), which vary during the El Niño events. The year-to-year variations of $NO_x$ emissions from lightning show large differences among CCMI models (Fig. S4), and only EMAC and GEOSCCM apply interactive soil $NO_x$ emissions that vary with meteorology conditions (Morgenstern et al., 2017) based on Yienger and Levy (1995). Thus $NO_x$ emissions from lightning and soil mainly contribute to inter-model differences instead of showing a consistent response to El Niño.

Using a machine learning method, Nicely et al. (2020) attributed the positive [OH] trend simulated by the CCMI models mainly to the increase of tropospheric $O_3$, $J(O^1D)$, $NO_x$ and $H_2O$, and attributed [OH] interannual variations to CO changes. Overall, the explanations of the drivers of OH year-to-year variations and trends found in our process analysis are broadly consistent with those reported by Nicely et al. (2020), and we emphasize that the decrease of CO emission and concentrations after 2000 (Zheng et al., 2019) is important for determining the accelerated positive OH trend.



### 3.3 Impact of OH variation on the top-down estimation of CH₄ budget

Fig.4a shows the anomaly of global total $CH_4$ emission estimated by inv_OH_std (nine scaled OH fields; yellow line) and inv_OH_cli (nine climatological OH; blue line) using the two-box model during 1986-2010. With the climatological OH fields (blue line), the top-down estimated $CH_4$ emissions show no clear trend before 2005, with large positive anomalies during strong El Niño years. There are two peaks of positive $CH_4$ emission anomalies during this period, 10.0Tg yr$^{-1}$ in 1991, and 14.4Tg yr$^{-1}$ in 1998. From 2005 to 2008, the $CH_4$ emissions show a large increase of 26.1Tg yr$^{-1}$.

The OH temporal variations are found to largely influence the interannual changes of top-down estimated $CH_4$ emissions (yellow line of Fig. 4a), with differences between the two inversions reaching up to more than 15Tg yr$^{-1}$ (Fig.4b). The contribution from the OH year-to-year variations and trends are also shown in Fig.4. The negative anomalies of OH during El Niño years reduce the unusually high top-down estimated $CH_4$ emissions in 1991-1992 by 6.7±2.6Tg yr$^{-1}$, and in 1998 by 9.7±3.1Tg yr$^{-1}$ (Fig.4c). As a result, the high emission peaks to match the observed $CH_4$ mixing ratio growth in 1991 (14.2ppb yr$^{-1}$) and 1998 (12.1ppbv yr$^{-1}$), as estimated using the climatological OH are largely reduced.

The identified positive OH trend leads to an additional 23.2±8.7Tg yr$^{-1}$ increase in $CH_4$ emissions from 1986 to 2010 (Fig.4d). During 1986-2005, the mean $CH_4$ emissions, as estimated with the scaled OH, show a positive trend of 0.63±0.43Tg yr$^{-2}$ (P<0.05). For 1986-2005, increased $CH_4$ emissions offset the increase in the OH sink to match the observations. From 2005 to 2008, in contrast to previous studies, which attribute the increased observed $CH_4$ mixing ratios to decreased OH based on MCF inversions (Turner et al., 2017, Rigby et al., 2017), the increasing OH trend simulated by CCMI models results in an additional 4.9±1.6Tg yr$^{-1}$ $CH_4$ emission increase in the inversion to match the observations.

We compare the inversion using the two-box model ("**x**" in Fig.5) with the results from the variational



approach (bars in Fig.5), using the multi-model mean OH field, to evaluate the performance of the simplified two-box model inversions. Despite the limitations inherent to two-box model inversions, such as treatment of inter-hemispheric transport, stratospheric loss, and the impact of spatial variability (Naus et al., 2019), the two-box model inversion estimates similar temporal changes of $CH_4$ emissions and losses compare to the variational approach for the four periods (Fig.5, left and middle), as well as their response to OH changes (Fig.5, right), on a global scale. Such comparisons reinforce the reliability of the conclusions made from the two-box model inversions regarding changes in the global total $CH_4$ budget.

The variational inversions allow us to access the regional contribution of the drivers to observed atmospheric $CH_4$ mixing ratio changes. Here, as a synthesis, we focus on four latitude bands (Fig.5 and Table S2), including the southern extra-tropical regions (90°S-30°S), the tropical regions (30°S-30°N), and the northern temperate (30°-60°N) and boreal (60°-90°N) regions. On average, OH over the tropical and northern temperate regions removes 74% and 14% of global total atmospheric $CH_4$, respectively.

Between the periods 1995-1996 and 1997-1998, if one does not consider the OH temporal variations (Inv_OH_cli), the $CH_4$ loss by OH shows a slight increase of 1.9Tg $yr^{-1}$ due to an increase of atmospheric $CH_4$ mixing ratios. The main driver of observed atmospheric $CH_4$ mixing ratio changes is the 9.5Tg $yr^{-1}$ increase of $CH_4$ emission over tropics and the 6.8Tg $yr^{-1}$ increase over the northern temperate regions (middle panel of Fig.5 and Table S2). When the multi-model mean OH temporal variations are included (Inv_OH_std), the negative anomaly of OH in 1997-1998 led to a 9.4Tg $yr^{-1}$ decrease in $CH_4$ loss in 1997-1998 compared to 1995-1996, of which 7.4Tg $y^{-1}$ (78%) are contributed by the tropical regions (left panel of Fig.5). As a result, the decrease of $CH_4$ loss by OH contributes a bit more to match the observed $CH_4$ mixing ratios increase during the El Niño periods than the changes in $CH_4$ emissions (a global increase of 8.2Tg $yr^{-1}$). The emission increases from 1995-1996 to 1997-1998 over the tropics and the northern temperate regions are reduced to 2.7Tg $yr^{-1}$ and 4.8Tg $yr^{-1}$ (left panel of Fig.5, Inv_OH_std), respectively,



similar to the inversion results given by Bousquet et al. (2006).


From the period 2001-2003 to 2007-2009, positive OH trends lead to a 13.0Tg yr$^{-1}$ increase of the CH$_4$ loss, of which 9.9Tg yr$^{-1}$ (76%) originates from the tropics (Inv_OH_std, left panel of Fig.5). In response to increased CH$_4$ losses, the increase of optimized emissions over tropical regions (15.7Tg yr$^{-1}$, Inv_OH_std) is more than twice of the inversion using climatological OH (7.3Tg yr$^{-1}$, Inv_OH_cli). The

emission increases during the two periods over the northern region show a smaller change of 2.0Tg yr$^{-1}$ (12.3 Tg yr$^{-1}$ estimated by Inv_OH_std versus 10.3 Tg yr$^{-1}$ by Inv_OH_cli, Fig. 5). The variational inversions show that the OH temporal variations are most critical for top-down estimates of CH$_4$ budgets over the tropical regions since OH over tropical regions shows larger interannual variations and trend than mid to high latitude regions (Fig.S5) and most of the CH$_4$ (74%) is removed from the atmosphere by OH

over the tropical regions.

## 4 Conclusion and discussion

Based on the simulations from the CCMI, we explore the response of OH fields to changes in climate, anthropogenic and natural emissions and its impact on the top-down estimates of CH$_4$ emissions during

1980-2010 based on a model perspective. We find that although CCMI models simulated rather different global total burdens of OH (Zhao et al., 2019), they show very similar patterns in temporal variations, including (1) negative anomalies during El Niño years, which are mainly driven by an elevated OH loss by reaction with CO from enhanced biomass burning, despite a partial buffering through enhanced OH production, and (2) a continuously increasing in OH from 1980, which is mostly contributed by OH

primary production and accelerating after 2000 due to reduced CO emissions. By conducting inversions using a two-box model and a variational approach together with the ensemble of CCMI OH fields, we find that (1) the OH year-to-year variations can largely reduce the CH$_4$ emission increase (by up to 10Tg yr$^{-1}$) needed to match the observed CH$_4$ increase during El Niño years, and (2) the positive OH trend



results in $23.2 \pm 8.7$ Tg yr$^{-1}$ additional increase in optimized emissions from 1986 to 2010 compared to the

inversions using constant OH. The variational inversions also show that OH temporal variations mainly

influence top-down estimates of CH$_4$ emissions over tropical regions.

The responses of OH to changes in biomass burning, ozone, water vapor, and lightning NO$_x$ emissions

during El Niño years have been recognized by previous studies (Holmes et al., 2013; Murray et al., 2014;

Turner et al., 2018; Rowlinson et al., 2019; Nguyen et al., 2020). Here, the consistent temporal variations

of CCMI OH fields increase our confidence in the model simulated response of OH to ENSO as a result

of several nonlinear chemical processes. One of the largest uncertainties is NO$_x$ emissions from lightning,

which have been proven to contribute to year-to-year variations in OH (Murray et al., 2013; Turner et al.,

2018), but here show a large spread among CCMI models. In addition, NO$_x$ emissions from soil may also

change during El Niño years. Improving estimates of NO$_x$ emissions from lightning based on satellite

observations (Murray et al., 2013) and a better representation of the interactive NO$_x$ emissions from the

soil are critical for improving the model simulation of OH temporal variability and for top-down estimates

of year-to-year variations of CH$_4$ emissions.

The positive trend of OH after the mid-2000s, which results in enhanced top-down estimated CH$_4$

emissions over the tropics, is opposite to those constrained by MCF inversions (Turner et al., 2017; Rigby

et al., 2017). However, the processes that control the model simulated positive OH trend are supported by

current studies based on observations, including decreased CO emissions (Zheng et al., 2019), small

variations of global NO$_x$ emissions (Miyazaki et al., 2017), and an increase in tropospheric ozone (Ziemke

et al., 2019) and water vapor (Chung et al., 2014). Given the large uncertainty existing in MCF-

constrained OH (Krol et al., 2003, Bousquet et al., 2005; Montzka et al., 2011; Prather and Holmes, 2017;

Naus et al. et al., 2019) and the evidence for increasing OH given by CCMI models and other literature,

the accuracy of MCF-based OH inversions after the mid-2000s remains an open problem and the large



discrepancy between MCF–based and CCMI model simulated OH trends requires more effort to close the gap.

The temporal variations of OH are generally less considered in current top-down estimates of $CH_4$ emissions, implying potential additional uncertainties in the global $CH_4$ budget (Saunois et al., 2017; Zhao et al., 2020). The tropical regions, where top-down estimated $CH_4$ emissions show the largest sensitivity to OH changes, represent more than 60% of $CH_4$ emissions worldwide (Saunois et al., 2016). The tropical $CH_4$ emissions are dominated by wetland emissions, of which large uncertainties exist in both bottom-up and top-down studies (Saunois et al., 2016; 2017). The variational inversions using OH with temporal variations attribute the observed rising $CH_4$ growth during El Niño to the reduction of $CH_4$ loss instead of enhanced emissions over tropics, which are consistent with process-based wetland models that estimated wetland $CH_4$ emission reductions at beginning of El Niño event (Hodson et al., 2011; Zhang et al., 2018). Future climate projections show that the extreme El Niño events will be more frequent under a warmer climate (Rao et al., 2019), which may enhance the fluctuations in [OH]. Furthermore, the changes in anthropogenic emissions, e.g. such as expected decreases in $NO_x$ emissions (Lamarque et al., 2013), can also affect the OH trends. Our research highlights the importance of considering climate changes and chemical feedbacks in future $CH_4$ budget research.

**Data availability**
The CCMI OH fields are available at the Centre for Environmental Data Analysis (CEDA; http://data.ceda.ac.uk/badc/wcrp-ccmi/data/CCMI-1/output; Hegglin and Lamarque, 2015), the Natural Environment Research Council's Data Repository for Atmospheric Science and Earth Observation. The CESM1-WACCM outputs for CCMI are available at http://www.earthsystemgrid.org (Climate Data Gateway at NCAR, 2019). The surface observations for $CH_4$ inversions are available at the World Data Centre for Greenhouse Gases (WDCGG, https://gaw.kishou.go.jp/, 2019). Other datasets can be accessed by contacting the corresponding author.



## Author contributions

YZ, BZ, MS, and PB designed the study, analyzed data and wrote the manuscript. AB developed the LMDz code for variational $CH_4$ inversions. ED provided the atmospheric in situ data. MH, MD, PJ, DK, OK, SS, and ST provided CCMI model outputs. All co-authors commented on the manuscript.

## Acknowledgements

This work benefited from the expertise of the Global Carbon Project methane initiative.

We acknowledge the modeling groups for making their simulations available for this analysis, the joint WCRP SPARC/IGAC Chemistry–Climate Model Initiative (CCMI) for organizing and coordinating the model simulations and data analysis activity, and the British Atmospheric Data Centre (BADC) for collecting and archiving the CCMI model output.

The EMAC model simulations have been performed at the German Climate Computing Centre (DKRZ) through support from the Bundesministerium für Bildung und Forschung (BMBF). DKRZ and its scientific steering committee are gratefully acknowledged for providing the HPC and data archiving resources for the consortial project ESCiMo (Earth System Chemistry integrated Modelling).

Makoto Deushi was partly supported by JSPS KAKENHI grant no. JP19K12312.

Yuanhong Zhao acknowledges helpful discussions with Zhen Zhang, Yilong Wang, and Lin Zhang.

## Competing interests

The authors declare that they have no conflicts of interest.

## Financial support

This research has been supported by the Gordon and Betty Moore Foundation (grant no. GBMF5439), "Advancing Understanding of the Global Methane Cycle".

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





**Figures & Tables**

**Table 1.** Two-box model inversion experiments.

| Inversion experiments | OH variability |
|---|---|
| Inv_OH_std | Full temporal changes (scaled OH fields) |
| Inv_OH_cli | Climatology OH (average of 1980-2010) |
| Inv_OH_var | Year-to-year variation only (detrend OH fields) |
| Inv_OH_trend | Trend only (remove OH year-to-year variation) |

**Table 2.** Multi-model mean $\pm$ standard deviation (SD) of annual total OH production (P) and loss (L) in Tmol yr$^{-1}$ and percentage contribution of each production and loss processes to total OH production and loss estimated with multi-model mean OH fields[1].

| Chemical reaction | Mean$\pm$SD | % |
|---|---|---|
| **Production** | 209.3$\pm$11.9 | / |
| $O(^1D)+H_2O$ | 96.2$\pm$1.9 | 46% |
| $NO+HO_2$ | 62.6$\pm$4.1 | 30% |
| $O_3+HO_2$ | 26.2$\pm$1.9 | 13% |
| **Other prod** | 24.2$\pm$7.0 | 12% |
| | | |
| **Loss**[1] | 209.3$\pm$11.9 | / |
| $CO+OH$ | 82.3$\pm$3.8 | 39% |
| $CH_4+OH$ | 32.4$\pm$1.4 | 15% |
| $CH_2O+OH$ | 11.5$\pm$0.5 | 6% |
| $Isoprene+OH$ | 13.1$\pm$1.4 | 6% |
| **Other loss** | 69.9$\pm$4.6 | 33% |

[1] The OH production and loss of the EMAC model are not included in the table since total OH production and loss are not given by the EMAC model.






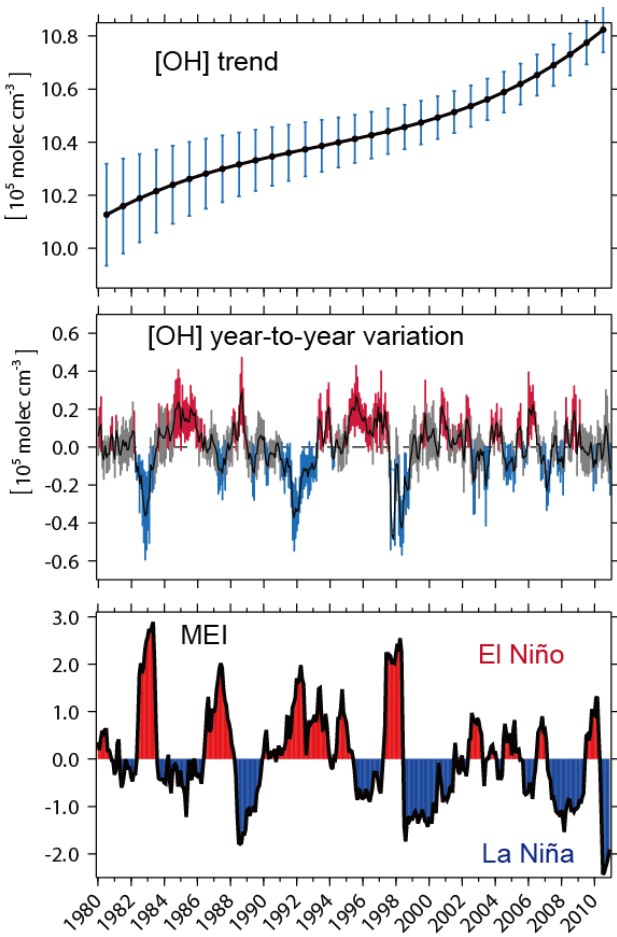

**Figure 1.** Top panel: Annual global tropospheric mean OH concentration ($[OH]_{GM-CH4}$, $CH_4$ reaction weighted) with year-to-year variations removed (represents the OH trend) simulated by CCMI models. The black line is the multi-model mean and associated error bars are standard deviations of different model results (also for the middle panel). Middle panel: Anomaly of detrended and deseasonalized monthly mean $[OH]_{GM-CH4}$ (represents the year-to-year variations of OH). Red bars indicate that the multi-model simulated $[OH]_{GM-CH4}$ are statistically significant ($P<0.05$) positive anomalies, blue bars indicate statistically significant negative anomalies, and grey bars indicate statistically non-significant anomalies. Bottom panel: Bi-monthly Multivariate ENSO Index (MEI).



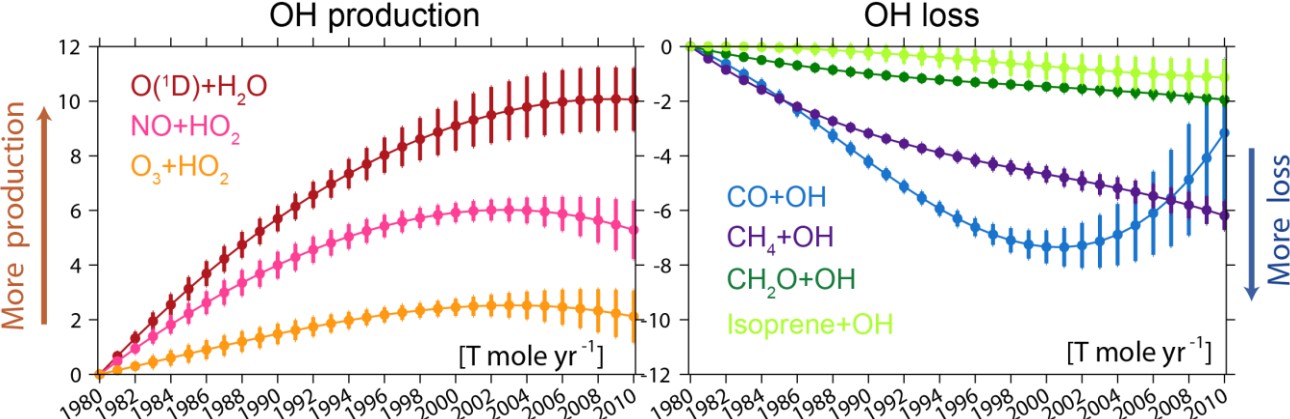

**Figure 2**. Annual total OH production and loss in Tmole yr$^{-1}$ with respect to the year 1980 with year-to-year variations removed.

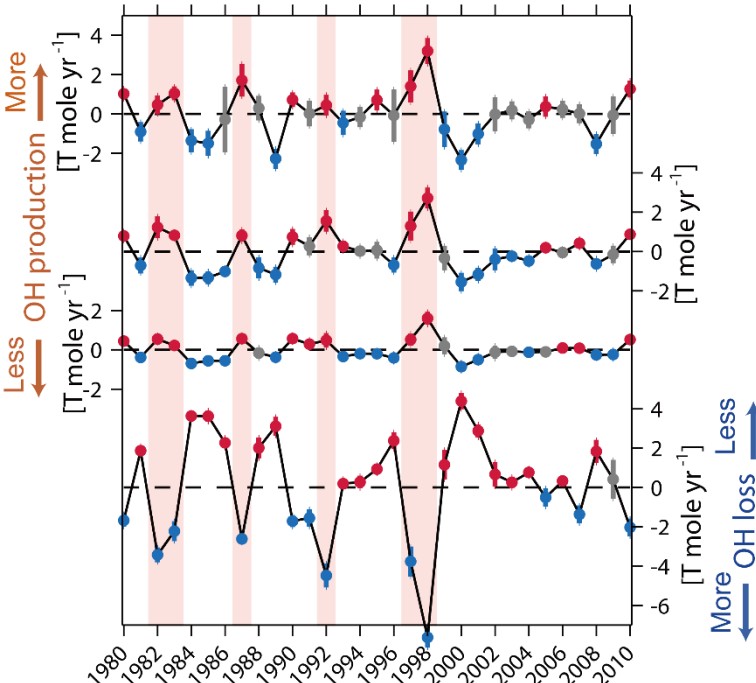

**Figure 3.** Anomaly of the detrended annual global total OH production from reactions O($^1$D)+H$_2$O, NO+HO$_2$, O$_3$+HO$_2$, and loss from reaction CO+OH. Black lines are multi-model means and the error bars are the standard deviations of all CCMI model results. The red, blue, and grey dots and error bars show statistically significant (P<0.05) positive anomalies, negative anomalies, and statistically non-significant anomalies, respectively. Shaded areas represent the El Niño years with more than 5 months of MEI>1.0.



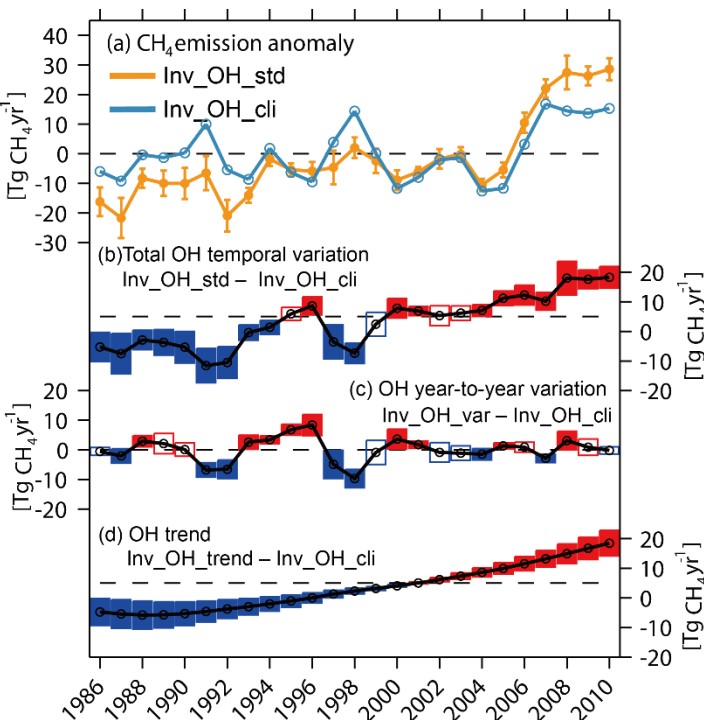

**Figure 4. (a)** Anomaly of global total CH$_4$ emissions using scaled CCMI OH fields (yellow line, Inv_OH_std), and climatological OH (blue, Inv_OH_cli) estimated by a two-box model inversion. The anomalies are calculated by comparing to the climatological mean CH$_4$ emissions of Inv_OH_cli over 1986-2010. **(b)** Influence of total OH temporal variations (OH year-to-year variation and trend, Inv_OH_std minus Inv_OH_cli), **(c)** OH year-to-year variations (Inv_OH_var minus Inv_OH_cli), and **(d)** OH trend (Inv_OH_trend minus Inv_OH_cli) on box-model estimated global total CH$_4$ emissions. The black lines are the mean of inversion results with different OH fields and the boxes are ±one standard deviation. The boxes with filled blue/red show OH lead to statistically significant (P<0.05) differences between the two inversions.





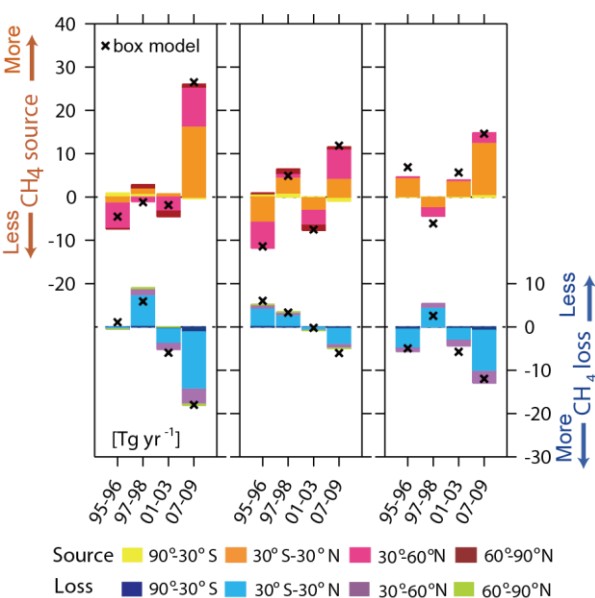

**Figure 5.** Anomaly of $CH_4$ emissions and losses estimated by variational 4D inversions (bars) and by two-box model inversions ("**x**") using a multi-model mean scaled OH (Inv_OH_std, left column) and climatological OH (middle column) during four time periods. The anomalies are calculated by comparing to the mean $CH_4$ emissions of Inv_OH_cli over the four time period. The differences between Inv_OH_std and Inv_OH_cli (Inv_OH_std minus Inv_OH_cli) are presented in the right column. The total emissions and loss over southern extra-tropical regions (90°S-30°S), the tropics (30°S-30°N), the northern temperate (30°-60°N), and the boreal (60°-90°N) regions are shown by different colors within each bar.