# Peer review of "On the role of trend and variability of hydroxyl radical (OH) in the global methane budget"

_Atmospheric Chemistry and Physics, 2020_

## Short Comment (SC1) · 10 May 2020

In line 204, "Of the total OH production, 46% (96.2±1.9Tg yr-1) ". The unit should be "Tmol yr-1"?
* * *

---

## Referee Comment (RC1) · Anonymous Referee #1 · 26 May 2020

The manuscript first discusses the OH variability and trends in details in relation with precursor emissions and chemistry as used in the chemistry-climate models. Then the modelled OH fields, adjusted to the global mean OH in 2000, are used for CH4 modelling in a box model and a sophisticated 3D model. They show good agreement (?) between the two models for global total CH4 budgets. The manuscript is well written and would be alright for publication in ACP. I would like to draw attention of the authors to a few points as detailed below. Hope these are useful.

line 49: Do you need an update here? Is the present day understanding is unambiguoustoo?

line 56ff: The citations look to be very restricted in general in these two para of the introduction. May consider expansion. The TransCom-CH4 project was launched to

underdstand the sources and sinks budget in comparison with the model transport, for example.

line 72ff: I am not aware of any proven issues with weakening MCF gradient. Could you pleas expand what exactly are you talking about; meridional, zonal or vertical gradients?

line 92: are CH4 budgets from 1980 or 1986

line 114: There are 4 issues with OH from CCMI models; you seems to ignore the biases in meridional gradient in OH, and account for the other three (global totals, trends and IAV)

line 120ff: Could this also mean that the variabilities you show are not from OH concentrations but due to t-dependent loss rates & dry airmass. Is it possible to tell the readers what would you expect if you scale OH concentrations themselves not weighted by k? including showing it in a 2nd column?

line 157ff: I fail to understand why continuous inversion were not done for the period 1994-2010, using the two OH cases. This does not seem to be for reducing computing time, given that 2 years are gone for spin-up and spin-down. Please explain. Also why you need two years of spin-up/down for the box model but only 1-year for the 3D model

line 185ff: What do you mean? I see many other negative anomalies are apparently consistent among the models.

line 195ff: I agree that the CH4 growth rates are more positive during the El Nino years (discussed in the TransCom-CH4 analysis too). We have to better understand the lower growth rates in CH4 during the La Nina periods - this is quite new concept. (some people talk about Mt Pinatubo for 1993 growth rate anomaly and others do not see a negative anomaly during the La Nina)

line 200: do you need "processes" here?

line 204: Is there a reason for different unit (Tg/yr) here?

line 212: should this be "different"? Do you mean the NMVOCs are not included in some of the model or the number of species differ from model to models?

line 214: My personal choice, but I would have loved to see the actual values presented in this plot. It is fine to adjust different multi-model values to a common 1980 level.

For here and elsewhere, this is specialised journal publication, there is no need for so much space restriction; I mean this can be 1-column figure is the trends are less prominent by the increase of y-axis range. Also for the x-axis tick labels, please consider reducing number of labels or elongate the x-axis or introduce minor ticks. Presently looks a bit clumsy

line 220ff: This is a nice discussion, but I cannot assess the novelty of it given that ACCMIP and CCMI paper have discussed the OH variabilities and budgets in similar fashion, and there are papers by MPI Mainz group on the details of OH budgets. Could you please consider showing the net (P-L) OH trends in a separate panel. When you say OH loss, is the '-ve' sign in the y-tick labels appropriate and consistent with the number in the text?

line 236ff: I was probably asking to present something similar in my earlier comment for OH anomaly in Fig. 1. May be it is good to show the Net OH (production - loss) variabilities as well in a separate panel here (in % change).

line 266ff: How good are the CO emission estimations and also the satellite data? I have heard some issues with the MOPITT data retrievals. Is this model comply with surface CO observations?

line 274ff: How good are these for accounting for the effect of the meteorology. I suppose the temperature effect is taken in to account by doing the k_oh+ch4 anomaly in Fig. 1, but there are likely to have some non-linear interaction between the transport (inter-hemispheric & stratosphere-troposphere exchange) and loss by OH. This effect

may be of 2nd order but nevertheless important. Any assessment would be helpful to the readers. This is where a long-term 3-D model based inversion would have helped.

line 280ff: It is obvious from this analysis that introducing OH IAV as modelled by the CCMI models will reduce the CH4 emission anomaly. What are the new questions/implications? 1) increase the wetland emission anomaly, 2) decrease biomass burning emission anomaly, 3) some missing process in the OH chemistry (recycling efficiency)

line 285ff: The most important question for the authors is then to convince the readers how you propose to increase CH4 emissions by more than 25 Tg/yr in just 3 years and keep maintaining at that level for the later years.

line 301: "assess"

line 303ff: Consider adding header to each panels - again panel size could be increased for clarity. It is very hard to see the semi-hemispheric emissions in the bars. The right panel adds to the confusion how to read this plot; for me it is much easy to see what you want say from the left and middle panels.

line 366: Did Prather and Holmes estimated OH variability or trends?

line 372: I do not know true or not true? The authors, I think, understand the trends and IAV in OH simulated by the CCMs still require much testing. Firstly the global mean OH values, which you have adjusted at the very first; the amplitude and phase of the modelled IAV may not be perfect; the longer term trends are still anyone's guess (needs at least one more line of evidence); finally there is unspoken bias in meridional gradients in the modelled OH. If a true variability and trend in OH become available there would be no issues with the top-down modellers to adopt it. There are several inversions which included the OH trends and variability like you discuss here (e.g., McNorton et al., ACP, 2018)

line 382: Is there a better reference for El Nino in future climate? I am sure there are

line 384: I think this is well known since the ACCMIP at the least!

---

## Referee Comment (RC2) · Anonymous Referee #2 · 19 Jun 2020

This study investigates the impact of CCM calculated OH fields on the long-term trend and interannual variability in global methane emissions as inferred from atmospheric inversions. It is concluded that accounting for the CCM simulated trend in OH implies a significantly larger increase in methane emissions in the past decades than previous estimates that did not consider changes in OH. In addition, correlations between OH variations and El Nino are found that reduce the variability in inferred emission, notably in the Tropics. The manuscript, that is well written, provides a useful contribution to the scientific discussion on the drivers of the global increase of methane, contrasting the view on the role of OH that follows from the more common use of MCF. The study also mentions important limitations of using MCF, suggesting that the CCM-derived OH scenario is more consistent. However, in my opinion the evidence in support of this

suggestion is very limited. Besides this point, there are a few other points that need further attention as explained below. Overall, only minor revisions will be needed to accept this paper for publication.

GENERAL COMMENTS

The study provides an analysis of the main drivers of variability and trends in OH using output from the CCMs. It is mentioned that both chemical and climatological influences are considered. However, the budget that is provided in Table 2 only lists chemical drivers. A study by Dentener et al (ACP, 1993) (which would be worth referencing in this context) identified an important contribution of meteorology on OH variability. Apart from changes in water vapour, those influences, including temperate and changes in circulating, are not discussed here.

The discussion of the results rightly mentions the different outcomes obtained using MCF or CCM derived OH, and their significance for global methane. From the evidence that is presented it is not possible in my opinion to conclude which of the views is right. Nevertheless, the conclusion section mentions that 'the evidence for increasing OH given by CCMI models and other literature, the accuracy of MCF-based OH inversions after the mid-2000s remains an open problem'. So, in short, the accuracy of MCF is the "open problem" that could explain the disagreement in the opinion of the authors. However, the consistency between CCM's itself cannot be considered as evidence, since these models are not independent and could therefore all be wrong for the same reason. To give an example that might introduce important uncertainty in CCM derived variations, the role of changes in aerosols on OH is not discussed at all. Either appropriate evidence should be presented of CCM's being more accurate than MCF or a more objective position should be taken regarding this question.

SPECIFIC COMMENTS

line 38, 40: the numbers that are mentioned in these sentences lack an uncertainty estimate.

line 148: How is Inv_OH_var detrended?

line 249: What is meant by 'response of OH to CO.'?

line 253: Banda et al 2016 is a good reference to add here for the effect of Mt. Pinatubo.

line 280-283: Similar conclusions on emission anomalies during El Nino were drawn by Butler et al, JGR, 2005, which would be useful references and comparing here.

Figure 3: what do the lines represent in this figure represent?

Figure 5: Emission anomalies are shown compared to what? Initially I assumed that the mean was subtracted. However, the bars for the different time slices don't add up to zero.

TECHNICAL CORRECTIONS

line 119: 'For each OH field' io ' .. fields'

line 122: The parenthesis in this sentence should be fixed.

line 170: 'This continuous increase in' io 'This continuously increases in', and 'based on MCF inversions' io 'based in the MCF inversions'

line 205: 'NO+HO2' io 'NO+NO2'

line 249: 'anomaly in OH primary' io 'anomaly OH primary'

line 250: 'during the 1991-1992..' io 'during 1991-1992..'

line 322: 'the Tropics' io 'the tropics'

line 324: 'twice that of the inversion' io 'twice of the inversion'

line 334: 'and their impact' io 'and its impact'
* * *

---

## Author Comment (AC1) · 14 Aug 2020

Comment: In line 204, "Of the total OH production, 46% (96.2±1.9Tg yr-1) ". The unit should be "Tmol yr-1"?

Response: We have changed the unit to "Tmol yr-1" in the text. Thank you very much for pointing out this.

---

## Author Comment (AC2) · 14 Aug 2020

*Reply to RC1: 'Review of "On the role of trend and variability of hydroxyl radical (OH) in the global methane budget"'*

**Comment:** The manuscript first discusses the OH variability and trends in details in relation with precursor emissions and chemistry as used in the chemistry-climate models. Then the modelled OH fields, adjusted to the global mean OH in 2000, are used for CH4 modelling in a box model and a sophisticated 3D model. They show good agreement (?) between the two models for global total CH4 budgets. The manuscript is well written and would be alright for publication in ACP. I would like to draw attention of the authors to a few points as detailed below. Hope these are useful.

**Response: We thank the reviewer for his/her helpful comments. All of them have been addressed in the revised manuscript. Please see our itemized responses below.**

**Comment: line 49**: Do you need an update here? Is the present day understanding is unambiguous too?

**Response: We change the sentence to (L47-49): "The tropospheric CH$_4$ mixing ratio has more than doubled between pre-industrial and the present day, mainly attributed to increasing anthropogenic CH$_4$ emissions (Etheridge et al., 1998; Turner et al. 2019)."**
**We add the reference: "Turner, A. J., Frankenberg, C., and Kort, E. A.: Interpreting contemporary trends in atmospheric methane, Proceedings of the National Academy of Sciences, 116, 2805-2813, 10.1073/pnas.1814297116, 2019."**

**Comment:** line 56: The citations look to be very restricted in general in these two para of the introduction. May consider expansion. The TransCom-CH4 project was launched to underdstand the sources and sinks budget in comparison with the model transport, for example.

**Response: We change the citation to "TransCom-CH4", thank you very much for pointing out this.**

**Comment: line 72:** I am not aware of any proven issues with weakening MCF gradient. Could you pleas expand what exactly are you talking about; meridional, zonal or vertical gradients?

**Response: We clarify in the text (L73-74):" ... and the weakening of inter-hemispheric MCF gradients after the 1990s (Krol et al., 2003, Bousquet et al., 2005; Montzka et al., 2011; Prather and Holmes, 2017)."**

**Comment:** line 92: are CH4 budgets from 1980 or 1986

**Response: We change the sentence to "... on decadal scale since the 1980s". and we**

add in L97-98:" We finally estimate the impact of OH year to year variations and trends on the top-down estimation of global CH$_4$ emissions over 1986-2010". Since we analyzed the OH variation for 1980-2010 and analyzed the inversion results for 1986-2010.

**Comment:** line 114: There are 4 issues with OH from CCMI models; you seems to ignore the biases in meridional gradient in OH, and account for the other three (global totals, trends and IAV)

**Response: In this work, we are focusing on the temporal variation of OH and the impact on CH$_4$. The impacts of OH inter-hemispheric gradient have been estimated in Zhao et al. (2020) and therefore less discussed in this study.**

**Reference:" Zhao, Y., Saunois, M., Bousquet, P., Lin, X., Berchet, A., Hegglin, M. I., Canadell, J. G., Jackson, R. B., Dlugokencky, E. J., Langenfelds, R. L., Ramonet, M., Worthy, D., and Zheng, B.: Influences of hydroxyl radicals (OH) on top-down estimates of the global and regional methane budgets, Atmos. Chem. Phys. Discuss., 2020, 1-45, 10.5194/acp-2019-1208, 2020."**

**Comment:** line 120: Could this also mean that the variabilities you show are not from OH concentrations but due to t-dependent loss rates & dry airmass. Is it possible to tell the readers what would you expect if you scale OH concentrations themselves not weighted by k? including showing it in a 2nd column?

**Response: For the scaling, we apply the single global scaling factor estimated for 2000 to every year of the OH field. Hence the scaling will not influence the interannual variation of OH. We have clarified this in the text (L116-117):" The inferred global mean scaling factors are calculated for the year 2000 and each OH field and then applied to the whole period (1980-2010)"**

**Comment:** line 157: I fail to understand why continuous inversion were not done for the period 1994-2010, using the two OH cases. This does not seem to be for reducing computing time, given that 2 years are gone for spin-up and spin-down. Please explain. Also why you need two years of spin-up/down for the box model but only 1-year for the 3D model.

**Response: We do the inversion separately for each time period so that the inversions can be run in parallel at the same time. The 3D model inversions are much more computationally expensive than the box model inversions. Hence we only take one-year spin-up and spin-down for the 3D model. We clarify in the text (L159-L160):" We only spin-up/spin-down the 4D variational inversions for one year to save computing time."**

**Comment:** line 185: What do you mean? I see many other negative anomalies are apparently consistent among the models.

**Response: We clarify this issue in the text by adding (L187-188):" Only the negative OH anomaly during 2006-2007 (2±1%) is simulated by all models during the four weak El Niño events."**

Comment: line 195ff: I agree that the CH4 growth rates are more positive during the El Nino years (discussed in the TransCom-CH4 analysis too). We have to better understand the lower growth rates in CH4 during the La Nina periods - this is quite new concept. (some people talk about Mt Pinatubo for 1993 growth rate anomaly and others do not see a negative anomaly during the La Nina).

**Response: Here we want to show that the CH$_4$ growth rate is smaller during the La Niña years comparing with their adjacent El Niño years. We make the expression more precise in the text (L96-98):" The positive anomalies of [OH]$_{GM-CH4}$ during La Niña events correspond to a much smaller CH$_4$ growth (e.g. 3.8±0.6ppbv yr$^{-1}$ in 1993 and 2.3±0.8ppbv yr$^{-1}$ in 1999) compared with that during the adjacent El Niño years (Fig. S1)."**

**Comment:** line 200: do you need "processes" here?

**Response: We remove the "processes".**

**Comment:** line 204: Is there a reason for different unit (Tg/yr) here?

**Response: We correct the typo by changing"Tg/yr" to "Tmol yr$^{-1}$"**

**Comment:** line 212: should this be "different"? Do you mean the NMVOCs are not included in some of the model or the number of species differ from model to models?

**Response: Here we mean the model outputs of OH loss due to reaction with NMVOCS, we clarify this in the text(L212-213):" Besides, there are 12% of OH production and 33% of OH loss not analyzed here due to lack of data in the CCMI model outputs (e.g. output of OH loss due to reaction with NMVOCs included in different models)"**

**Comment:** line 214: My personal choice, but I would have loved to see the actual values presented in this plot. It is fine to adjust different multi-model values to a common 1980 level.

**Response: We agree that it will be more straightforward to show the actual values. However, we here adjust the model values to a common 1980 to focus more on the temporal variations from 1980.**

**Comment:** For here and elsewhere, this is specialised journal publication, there is no need for so much space restriction; I mean this can be 1-column figure is the trends are less prominent by the increase of y-axis range. Also for the x-axis tick labels, please consider reducing number of labels or elongate the x-axis or introduce minor ticks. Presently looks a bit clumsy.

**Response: We plot the trends of different chemical processes in the same panel to better compare the relative contribution from each process and find out which process is most important for determining OH trend. We replot figure 1-4 to reduce x-axis tick labels as suggested.**

**Comment:** line 220: This is a nice discussion, but I cannot assess the novelty of it given that ACCMIP and CCMI paper have discussed the OH variabilities and budgets in similar fashion, and there are papers by MPI Mainz group on the details of OH budgets. Could you please consider showing the net (P-L) OH trends in a separate panel. When you say OH loss, is the '-ve' sign in the y-tick labels appropriate and consistent with the number in the text?

**Response: ACCMIP provides the model outputs for time slices (Naik et al., 2013), which limit the analysis of OH interannual variations. For the studies based on CCMI model simulations, both Zhao et al. (2019) and Nicely et al. (2020) analyzed the OH interannual variations but not the OH budget. The OH budget has already been analyzed in previous single model studies such as Murray et al. (2013; 2014) (as we cite in the manuscript) and Lelieveld et al. (2016) (paper from MPI Mainz group). But to our knowledge, few studies are analyzing the interannual variation of OH budget based on multi-model outputs.**

**We cite Leliveld et al. (2016) in the text (L201-203): "Here we assess the drivers of OH year-to-year variations and trend by calculating the OH production and loss processes listed in Table 2 following Murray et al. (2013; 2014) and Lelieveld et al. (2016)."**

**Reference:" Lelieveld, J., Gromov, S., Pozzer, A., and Taraborrelli, D.: Global tropospheric hydroxyl distribution, budget and reactivity, Atmos. Chem. Phys., 16, 12477-12493, 10.5194/acp-16-12477-2016, 2016."**

**Showing the net (P-L) will certainly help for better understanding the OH variations. However, only 2 of 5 models provide both total OH production and OH loss data, so we cannot give the temporal variations of net OH production and loss based on current model outputs.**

For the "-ve" sign (I suppose the reviewer means "negative"), we now clarify this issue in the figure caption.

"Figure 2. Annual total OH tendency (Tmole yr$^{-1}$) from chemical reactions with respect to the year 1980 with year-to-year variations removed. The positive and negative tendencies represent OH production (left) and loss processes (right), respectively."

"Figure 3. Anomaly of the detrended annual global total OH tendency from reactions O($^1$D)+H$_2$O, NO+HO$_2$, O$_3$+HO$_2$, and CO+OH. The positive and negative tendencies represent OH production and loss processes, respectively."

Comment: line 236: I was probably asking to present something similar in my earlier comment for OH anomaly in Fig. 1. May be it is good to show the Net OH (production - loss) variabilities as well in a separate panel here (in % change).

Response: As stated in response to the last comments, we cannot assess the net OH production and loss for most of the models due to a lack of corresponding output.

Comment: line 266: How good are the CO emission estimations and also the satellite data? I have heard some issues with the MOPITT data retrievals. Is this model comply with surface CO observations?

Response: The trend and variations of the tropospheric CO column simulated by CCMI models have been evaluated by comparison with MOPITT data retrievals (Strode et al., 2016). Here we are focusing on the OH loss by CO over the whole troposphere (instead of the surface). The consistency of CCMI simulated tropospheric CO column with MOPITT observations can support the model simulated decreasing tropospheric OH loss by reaction with CO during 2000-2010.

We add in the text (L220-222): "The negative trend of CO simulated by CCMI models during 2000-2010 is consistent with MOPITT observations over most of the regions (Strode et al., 2016)."

Despite the finding that atmospheric chemistry models generally capture the CO trend, they usually underestimate the atmospheric CO burden compared to surface and satellite observations. We add in the text (381-383): "For example, underestimation of CO, especially over the northern hemisphere, compared with the surface and satellite observations (Naik et al., 2013; Strode et al., 2016) and bias in atmospheric O$_3$ column (Zhao et al. 2019)."

We add the reference:" Strode, S. A., Worden, H. M., Damon, M., Douglass, A. R., Duncan, B. N., Emmons, L. K., Lamarque, J. F., Manyin, M., Oman, L. D., Rodriguez, J. M., Strahan, S. E., and Tilmes, S.: Interpreting space-based trends

in carbon monoxide with multiple models, Atmos. Chem. Phys., 16, 7285-7294, 10.5194/acp-16-7285-2016, 2016."

Comment: line 274: How good are these for accounting for the effect of the meteorology. I suppose the temperature effect is taken in to account by doing the k_oh+ch4 anomalyin Fig. 1, but there are likely to have some non-linear interaction between the transport (inter-hemispheric & stratosphere-troposphere exchange) and loss by OH. This effect may be of 2nd order but nevertheless important. Any assessment would be helpful to the readers. This is where a long-term 3-D model based inversion would have helped.

**Response: The long-term 3D model inversions can certainly help better assess the impact of OH variations. However, the 3D model inversions are computationally expensive, which limits conducting long-term 3D model inversions using all of the 9 OH fields present in this study. That's why we use the two-box model to do long-term inversions and do 3D model inversions with a focus on four time periods.**

**We have demonstrated the advantage of both box model and 3D model inversions in the text (L130-134): "The two-box model inversions allow us to easily conduct multiple long-term global scale inversions (1984-2012) with each of the nine OH fields to estimate the global CH$_4$ emission variations caused by various OH fields. The 4D variational inversions allow us to better represent atmospheric transport, account for the variation of meteorological conditions, and address regional CH$_4$ emission distributions."**

**And we have compared the emission changes estimated by two-box model inversions and 3D model inversion in Figure 5, which show that (L302-L306) "Despite the limitations inherent to two-box model inversions, such as treatment of inter-hemispheric transport, stratospheric loss, and the impact of spatial variability (Naus et al., 2019), the two-box model inversion estimates similar temporal changes of CH$_4$ emissions and losses compare to the variational approach for the four periods, as well as their response to OH changes (Fig.5), on a global scale. Such comparisons reinforce the reliability of the conclusions made from the two-box model inversions regarding changes in the global total CH$_4$ budget.".**

Line280 : It is obvious from this analysis that introducing OH IAV as modelled by the CCMI models will reduce the CH4 emission anomaly. What are the new questions/implications? 1) increase the wetland emission anomaly, 2) decrease biomass burning emission anomaly, 3) some missing process in the OH chemistry (recycling efficiency).

**Response: We have discussed the implication on top-down estimated wetland**

emissions in the "Conclusion and discussion" (L393-396): "The variational inversions using OH with temporal variations attribute the observed rising CH$_4$ growth during El Niño to the reduction of CH$_4$ loss instead of enhanced emissions over the tropics, which are consistent with process-based wetland models that estimated wetland CH$_4$ emission reductions at beginning of El Niño event (Hodson et al., 2011; Zhang et al., 2018)."

We also discuss the impact on top-down estimated biomass burning emissions (L400-L401): "Also, the negative OH anomaly can reduce the top-down estimated biomass burning CH$_4$ emission spike during El Niño events, similar to that presented in Bousquet et al. (2006)."

We discuss the implications of including full-chemistry estimated OH by comparing with previous results which only consider OH loss by CO (L362-365): "We estimated that the negative OH anomaly in 1998 reduces the high top-down estimated CH$_4$ emissions by 10±3Tg yr$^{-1}$, ~40% smaller than the reduction estimated by Butler et al. (2005) (16Tg yr$^{-1}$), which only include the OH reduction response to enhanced biomass burning CO emissions. The smaller CH$_4$ emission reductions (OH anomalies) estimated with CCMI OH fields shows the significance of considering multi chemical processes as included in the 3D atmospheric chemistry model in capturing OH variations and inverting for CH$_4$ emissions"

Reference: "Butler, T. M., Rayner, P. J., Simmonds, I., and Lawrence, M. G.: Simultaneous mass balance inverse modeling of methane and carbon monoxide, Journal of Geophysical Research: Atmospheres, 110, 10.1029/2005jd006071, 2005."

**Comment:** line 285: The most important question for the authors is then to convince the readers how you propose to increase CH4 emissions by more than 25 Tg/yr in just 3 years and keep maintaining at that level for the later years.

Response: The large emission increase after 2005 have been reported in previous studies (e.g. Kirschke et al. (2013), Saunois et al. (2017)). We compare the emission increase over the same period with previous study in the text (L279-281):" The CH$_4$ emissions averaged over 2006-2010 is 20Tg yr$^{-1}$ higher than over 2000-2005, within the range of 17–22Tg yr$^{-1}$ estimated by an ensemble of inversions in Kirschke et al. (2013)."

We add the reference: " Kirschke, S., Bousquet, P., Ciais, P., Saunois, M., Canadell, J. G., Dlugokencky, E. J., Bergamaschi, P., Bergmann, D., Blake, D. R., Bruhwiler, L., Cameron-Smith, P., Castaldi, S., Chevallier, F., Feng, L., Fraser, A., Heimann, M., Hodson, E. L., Houweling, S., Josse, B., Fraser, P. J., Krummel, P. B., Lamarque, J.-F., Langenfelds, R. L., Le Quéré, C., Naik, V., O'Doherty, S., Palmer, P. I., Pison, I., Plummer, D., Poulter, B., Prinn, R. G., Rigby, M., Ringeval, B., Santini, M., Schmidt, M., Shindell, D. T., Simpson, I. J., Spahni, R., Steele, L. P.,

**Strode, S. A., Sudo, K., Szopa, S., van der Werf, G. R., Voulgarakis, A., van Weele, M., Weiss, R. F., Williams, J. E., and Zeng, G.: Three decades of global methane sources and sinks, Nature Geoscience, 6, 813-823, https://doi.org/10.1038/ngeo1955, 2013."**

**Comment:** line 301: "assess"
**Response: Changes as suggested**

line 303ff: Consider adding header to each panels - again panel size could be increased for clarity. It is very hard to see the semi-hemispheric emissions in the bars. The right panel adds to the confusion how to read this plot; for me it is much easy to see what you want say from the left and middle panels.

**Response: We change the figure as suggested.**

[Figure]

**Comment**: line 366: Did Prather and Holmes estimated OH variability or trends?
**Response: Prather and Holmes (2017) showed uncertainties exist in the MCF-constrained OH using two-box models.**

**Comment:** line 372: I do not know true or not true? The authors, I think, understand the trends and IAV in OH simulated by the CCMs still require much testing. Firstly the global mean OH values, which you have adjusted at the very first; the amplitude and phase of the modelled IAV may not be perfect; the longer term trends are still anyone's guess (needs at least one more line of evidence); finally there is unspoken bias in meridional gradients in the modelled OH. If a true variability and trend in OH become available there would be no issues with the top-down modellers to adopt it. There are several inversions which included the OH trends and variability like you discuss here

(e.g., McNorton et al., ACP, 2018)

**Response: We discuss the uncertainties in OH simulated by atmospheric chemistry models in the text now in more detail: "However, the CCMI models still show biases that are related to OH production and loss. For example, these include an underestimation of CO especially over the northern hemisphere compared with the surface and satellite observations (Naik et al., 2013; Strode et al., 2016) and bias in atmospheric total $O_3$ column (Zhao et al. 2019). In addition, the changes in aerosols (Tang et al., 2003) and atmospheric circulation such as the Hadley cell expansion (Nicely et al., 2018) are not discussed in this study. Given the large discrepancy between MCF–based and CCMI model simulated OH trends, and the uncertainties in both model simulated (Naik et al., 2013; Zhao et al., 2019) and MCF-constrained (Bousquet et al., 2005; Prather and Holmes, 2017; Naus et al. et al., 2019) OH, the OH trend after the mid-2000s remains an open problem and more effort is required in both method to explore the close the gap. "**

**We change L372 to: "The temporal variations of OH, which are generally not well constrained in current top-down estimates of $CH_4$ emissions, imply potential additional uncertainties in the global $CH_4$ budget (Saunois et al., 2017; Zhao et al., 2020)."**

**Comment:** line 382: Is there a better reference for El Nino in future climate? I am sure there are

**Response: We change the reference to "Berner, J., Christensen, H. M., and Sardeshmukh, P. D.: Does ENSO Regularity Increase in a Warming Climate?, Journal of Climate, 33, 1247-1259, 10.1175/jcli-d-19-0545.1, 2020."**

**Comment:** line 384: I think this is well known since the ACCMIP at the least!

**Response: We change the sentence to "Our research emphasizes the importance of considering climate changes and chemical feedbacks related to OH in future $CH_4$ budget research."**

---

## Author Comment (AC3) · 14 Aug 2020

*Reply to RC2: 'Referee comments'*

This study investigates the impact of CCM calculated OH fields on the long-term trend and interannual variability in global methane emissions as inferred from atmospheric inversions. It is concluded that accounting for the CCM simulated trend in OH implies a significantly larger increase in methane emissions in the past decades than previous estimates that did not consider changes in OH. In addition, correlations between OH variations and El Nino are found that reduce the variability in inferred emission, notably in the Tropics. The manuscript, that is well written, provides a useful contribution to the scientific discussion on the drivers of the global increase of methane, contrasting the view on the role of OH that follows from the more common use of MCF. The study also mentions important limitations of using MCF, suggesting that the CCM-derived OH scenario is more consistent. However, in my opinion the evidence in support of this suggestion is very limited. Besides this point, there are a few other points that need further attention as explained below. Overall, only minor revisions will be needed to accept this paper for publication.

**Response: We thank the reviewer for his/her helpful comments. We have now discussed the limitations of the CCMI-derived OH in more detail in the text. All of the other comments have been addressed in the revised manuscript. Please see the itemized responses below.**

**Comment:** The study provides an analysis of the main drivers of variability and trends in OH using output from the CCMs. It is mentioned that both chemical and climatological influences are considered. However, the budget that is provided in Table 2 only lists chemical drivers. A study by Dentener et al (ACP, 1993) (which would be worth referencing in this context) identified an important contribution of meteorology on OH variability. Apart from changes in water vapour, those influences, including temperate and changes in circulating, are not discussed here.

**Response: Dentener et al. showed that water vapor is the main meteorological driver of OH trend during 1979-1993. Our analysis also shows that the increase of water vapor can contribute to a positive OH trend by enhancing OH primary production (L217-L219): "The increase in OH primary production is due to an increase in both tropospheric $O_3$ burden (producing $O(^1D)$) and water vapor (Dentener et al 2003; Zhao et al., 2019; Nicely et al., 2020)."**

**The impact of water vapor on OH variability is also discussed in the manuscript (L246-L248): "The increase in OH primary production is mainly due to an increase in tropospheric water vapor and $O_3$ burden during El Niño events (Fig.S3 and S12 in Nicely et al. (2020)), while the increase in OH secondary production is caused by enhanced $NO_x$ emissions (Fig.S3) and $O_3$ formation (Nicely et al. (2020) related to biomass burning as well as more $HO_2$ formation by CO+OH."**

The temperature mainly influences the reaction rates of OH with other chemical species (CO, CH4, NMVOC, etc.), which is included in our estimation of OH production and loss (Table S1). Nicely et al. (2018; 2020) have proven that temperature has a neglectable impact on OH trend.

Due to a short lifetime, the OH is controlled by local production and loss, thus less directly impacted by atmospheric transport. Nicely et al. (2018) shown that the Hadley cell expansion can have a small impact on tropospheric mean [OH] through changing tropopause height. We add this point in the text (L380-383): "In addition, the changes in aerosols (Tang et al., 2003) and atmospheric circulation such as Hadley cell expansion (Nicely et al., 2018) are not discussed in this study."

We also add the following references:
"Dentener, F., Peters, W., Krol, M., van Weele, M., Bergamaschi, P., and Lelieveld, J.: Interannual variability and trend of CH4 lifetime as a measure for OH changes in the 1979–1993 time period, Journal of Geophysical Research: Atmospheres, 108, 4442, 10.1029/2002jd002916, 2003."
" Nicely, J. M., Canty, T. P., Manyin, M., Oman, L. D., Salawitch, R. J., Steenrod, S. D., Strahan, S. E., and Strode, S. A.: Changes in Global Tropospheric OH Expected as a Result of Climate Change Over the Last Several Decades, Journal of Geophysical Research: Atmospheres, 123, 10,774-710,795, doi:10.1029/2018JD028388, 2018."

Comment: The discussion of the results rightly mentions the different outcomes obtained using MCF or CCM derived OH, and their significance for global methane. From the evidence that is presented it is not possible in my opinion to conclude which of the views is right. Nevertheless, the conclusion section mentions that 'the evidence for increasing OH given by CCMI models and other literature, the accuracy of MCF-based OH inversions after the mid-2000s remains an open problem'. So, in short, the accuracy of MCF is the "open problem" that could explain the disagreement in the opinion of the authors. However, the consistency between CCM's itself cannot be considered as evidence, since these models are not independent and could therefore all be wrong for the same reason. To give an example that might introduce important uncertainty in CCM derived variations, the role of changes in aerosols on OH is not discussed at all. Either appropriate evidence should be presented of CCM's being more accurate than MCF or a more objective position should be taken regarding this question.

Response: We try to discuss the two methods in the text now in a more balanced way:

"However, the CCMI models still show biases that related to OH production and loss. For example, underestimation of CO especially over the northern hemisphere compared with the surface and satellite observations (Naik et al., 2013; Strode et al., 2016) and bias in atmospheric total O₃ column (Zhao et al. 2019). In addition,

the changes in aerosols (Tang et al., 2003) and atmospheric circulation such as Hadley cell expansion (Nicely et al., 2018) are not discussed in this study. Given the uncertainties in both atmospheric chemistry model simulated (Naik et al., 2013; Zhao et al., 2019) and MCF-constrained OH (Bousquet et al., 2005; Prather and Holmes, 2017; Naus et al. et al., 2019), and the large discrepancy between two methods, the OH trend after the mid-2000s remains an open problem and more effort is required in both method to close the gap. "

We add the references:
"Strode, S. A., Duncan, B. N., Yegorova, E. A., Kouatchou, J., Ziemke, J. R., and Douglass, A. R.: Implications of carbon monoxide bias for methane lifetime and atmospheric composition in chemistry climate models, Atmos. Chem. Phys., 15, 11789-11805, 10.5194/acp-15-11789-2015, 2015."
" Tang, Y., Carmichael, G. R., Uno, I., Woo, J.-H., Kurata, G., Lefer, B., Shetter, R. E., Huang, H., Anderson, B. E., Avery, M. A., Clarke, A. D., and Blake, D. R.: Impacts of aerosols and clouds on photolysis frequencies and photochemistry during TRACE-P: 2. Three-dimensional study using a regional chemical transport model, Journal of Geophysical Research: Atmospheres, 108, 8822, 10.1029/2002jd003100, 2003."

SPECIFIC COMMENTS
**Comment:** line 38, 40: the numbers that are mentioned in these sentences lack an uncertainty estimate.

**Response: We change "up to 10Tg yr$^{-1}$" to "up to 10±3Tg yr$^{-1}$" and "increase by 23Tg yr$^{-1}$" to "increase by 23±9Tg yr$^{-1}$".**

**Comment:** line 148: How is Inv_OH_var detrended?

**Response: We clarify in the text "Inv_OH_var stands for the inversion using the detrended OH".**

**Comment:** line 249: What is meant by 'response of OH to CO.'?

**Response: We clarify by changing the "response of OH to CO" to "response of OH to enhanced CO emissions during the El Niño events".**

**Comment**: line 253: Banda et al 2016 is a good reference to add here for the effect of Mt. Pinatubo.

**Response: Thank you very much for providing this valuable reference. We revise the sentence to:" The positive anomaly OH primary production (0.2±0.5Tmol yr$^{-1}$) is not significant during 1991-1992 El Niño event, maybe due to absorption of ultraviolet (UV) by volcanic SO₂ and scattering of UV by sulfate aerosols as well**

**as reduction of tropospheric water vapor after the eruption of Mount Pinatubo (Bândă et al., 2016; Soden et al., 2020).”**

**We add reference:” Bândă, N., Krol, M., van Weele, M., van Noije, T., Le Sager, P., and Röckmann, T.: Can we explain the observed methane variability after the Mount Pinatubo eruption?, Atmos. Chem. Phys., 16, 195-214, 10.5194/acp-16-195-2016, 2016.”**

**Comments:** line 280-283: Similar conclusions on emission anomalies during El Nino were drawn by Butler et al, JGR, 2005, which would be useful references and comparing here.

**Response: we add in the text (L358-L363): ” We estimated that the negative OH anomaly in 1998 reduces the high top-down estimated CH4 emissions by 10±3Tg yr$^{-1}$, ~40% smaller than the reduction estimated by Butler et al. (2005) (16Tg yr$^{-1}$), which only include the OH reduction response to enhanced biomass burning CO emissions. The smaller CH4 emission reduction (OH anomaly) estimated with CCMI OH fields may reflect the significance of considering multi chemical processes as included in the 3D atmospheric chemistry model in capturing OH variations and inversing CH4 emissions.”**

**Comment:** Figure 3: what do the lines represent in this figure represent? Figure 5: Emission anomalies are shown compared to what? Initially I assumed that the mean was subtracted. However, the bars for the different time slices don’t add up to zero.

**Response: we add the legend in Figure 3 as shown below. The anomalies in Figure 5 are calculated by comparing to the mean CH4 emissions of Inv_OH_cli over the four time period as demonstrated in the figure caption, we add the value “494Tg” in the figure caption to clarify.**

[Figure]

TECHNICAL CORRECTIONS
**Comment:** line 119: 'For each OH field' io ' .. fields'

**Response: Changed as suggested**

**Comment:** line 122: The parenthesis in this sentence should be fixed.

**Response: We change the parenthesis to "($[OH]_{GM-CH4}$, weighting factor = reaction rate of OH with $CH_4$ $\times$ dry air mass, Lawrence et al., 2001) as well as of its production and loss rates."**

**Comments:**
line 170: 'This continuous increase in' io 'This continuously increases in', and 'based on MCF inversions' io 'based in the MCF inversions'
line 205: 'NO+HO2' io 'NO+NO2'
line 249: 'anomaly in OH primary' io 'anomaly OH primary'
line 250: 'during the 1991-1992..' io 'during 1991-1992..'
line 322: 'the Tropics' io 'the tropics'
line 324: 'twice that of the inversion' io 'twice of the inversion'
line 334: 'and their impact' io 'and its impact'

**Response: The above technical corrections are changes as suggested. Thank you very much for pointing out these.**